# Changes in anxiety in the general population over a six-year period

**Andreas Hinz**[1]*, **Peter Esser**[1], **Michael Friedrich**[1], **Heide Glaesmer**[1], **Anja Mehnert-Theuerkauf**[1], **Matthias L. Schroeter**[2,3], **Katja Petrowski**[4,5], **Anne Toussaint**[6]

**1** Department of Medical Psychology and Medical Sociology, University of Leipzig, Leipzig, Germany, **2** Max Planck Institute for Human Cognitive and Brain Sciences, Leipzig & Clinic for Cognitive Neurology, University Hospital Leipzig, Leipzig, Germany, **3** Clinic for Cognitive Neurology, University Hospital Leipzig, Leipzig, Germany, **4** Medical Psychology & Medical Sociology, University Medical Center of the Johannes Gutenberg University Mainz, Mainz, Germany, **5** Department of Internal Medicine III, Dresden University of Technology, Dresden, Germany, **6** Department of Psychosomatic Medicine and Psychotherapy, University Medical Center Hamburg-Eppendorf, Hamburg, Germany

\* andreas.hinz@medizin.uni-leipzig.de

## Abstract

### Background

Anxiety is a frequent condition in patients and in the general population. The aim of this study was to investigate changes in anxiety over time and to test several psychometric properties of the Generalized Anxiety Disorder Screener (GAD-7) from a longitudinal perspective.

### Methods

The GAD-7 was included in an examination with two waves, six years apart. The study sample ($n = 5355$) was comprised of representatively selected adults from the general population with a mean age of 57.3 (SD = 12.3) years.

### Results

During the 6-year time interval, anxiety increased significantly from 3.28 ± 3.16 (t1) to 3.66 ± 3.46 (t2). Confirmatory factor analyses proved the longitudinal measurement invariance of the GAD-7. Reliability of the GAD-7 was established both for the cross-sectional and the longitudinal perspective. The test-retest correlation was $r = 0.53$, and there were no substantial sex or age differences in these coefficients of temporal stability. The mean changes in anxiety were similar for males and females, and there was no linear age trend in the changes measured by the GAD-7. Changes in anxiety over the 6-year period were correlated with changes in satisfaction with life ($r = -0.30$), bodily complaints ($r = 0.31$), and the mental component of quality of life ($r = -0.48$).

### Conclusion

The GAD-7 is a suitable instrument for measuring changes in anxiety. Age and gender have only minor significance when interpreting change scores.

**Data Availability Statement:** Data cannot be shared publicly because of data protection reasons. Data availability is only possible via a project agreement. Data are available from the LIFE-Adult-Study Institutional Data Access / Ethics Committee

(contact via https://ldp.life.uni-leipzig.de/) for researchers who meet the criteria for access to confidential data.

**Funding:** The author(s) received no specific funding for this work.

**Competing interests:** The authors have declared that no competing interests exist.

## Introduction

Anxiety disorders are common in patients and in the general population [1]. In primary care settings, anxiety disorders are among the most frequent disorders observed [2–4], and these disorders are associated with high use of health care services [5]. Sex and age differences in anxiety have been investigated in several studies. Females generally report higher degrees of anxiety than men do [6–8], while age differences in anxiety are less clear. Most studies found nonlinear and unsystematic effects of age on anxiety [7, 9, 10].

While there are multiple cross-sectional studies on anxiety with samples of the general population, longitudinal studies are rare. However, to interpret changes in anxiety over time in patients, it is relevant to know which changes occur in the general population. A related question concerns the temporal stability of anxiety. A compilation of several studies on test-retest correlations of anxiety and other variables of mental health [11] showed coefficients between 0.55 and 0.70, with decreasing coefficients with increasing temporal distance between the measurements. However, the question of how anxiety's temporal stability of anxiety depends on sociodemographic factors, e.g., whether women are more or less constant in their anxiety level than men, and how the stability of anxiety depends on age or socioeconomic level, had yet to be systematically tested.

Multiple studies have investigated the (cross-sectional) association between anxiety and other variables such as depression [12], physical complaints [13], fatigue [14, 15], fear of cancer progression [16] and COVID-19 risk perception [17]. Such studies are useful for clarifying the partial overlap between related constructs and symptoms. However, from a longitudinal perspective it is also of interest whether changes in anxiety correspond with changes in those variables. There are only a few studies that have investigated the associations between change scores of mental health variables [18].

The GAD-7 [19] is a frequently used tool for measuring generalized anxiety. This questionnaire has been translated into multiple languages and has been validated for several clinical groups. Normative values are available [6], and multiple studies have proved the psychometric quality of the GAD-7 [6, 20–23] from a cross-sectional perspective. A further issue of the longitudinal psychometric quality of a questionnaire is reliability of change. The common reliability in terms of Cronbach's alpha is high when the items of the questionnaire are highly correlated with each other. This cross-sectional view on reliability can be applied to the longitudinal analysis: To what degree are the changes of the items intercorrelated, and how do the changes in the items contribute to the total change score? This type of reliability analysis will also be presented here.

The GAD-2 is an ultra-short form of the GAD-7. It consists of the two psychometrically most reliable items of the GAD-7 [12, 24]. Because there is a need for very brief instruments that can be effectively used in epidemiological research, it is also relevant to test the cross-sectional and longitudinal psychometric properties of this ultra-short instrument.

Measurement invariance of the GAD-7 across sex and age has been tested in several cross-sectional studies with samples of the general population and clinical samples [6, 7, 25, 26], and longitudinal measurement invariance across several time points has been examined in certain groups of patients [27, 28]. Such analyses had yet to be performed with samples of the general population though.

Based on the data of this follow-up study, the aims of this paper were

(a) to analyze changes in anxiety during a 6-year interval, (b) to test psychometric properties of the GAD-7 and the GAD-2 including coefficients of temporal stability, reliability of change, and measurement invariance, (c) to analyze sex and age differences in the changes of anxiety, and (d) to examine the associations between anxiety and other variables (quality of

life, bodily complaints, life satisfaction, habitual optimism, and social support) both from a cross-sectional and from a longitudinal perspective.

## Material and methods

### Sample

The LIFE-Adult-Study of the Leipzig Center for Civilization Diseases (LIFE) is a population-based study with a representative sample ($n$ = 10000) of people living in the city of Leipzig, Germany. The first wave of this study was conducted between 2011 and 2014. The local residents' registration office generated an age- and gender- stratified random selection of inhabitants, ranging in age from 18 to 80 years. According to the study protocol, the focus was on the age range 40–80 years. At the study center, the study participants underwent a sequence of assessments, including collection of their sociodemographic data, behavioral and lifestyle factors, medical history, and several medical examinations. Details of the study design have been published elsewhere [29].

Between 2017 and 2021, all participants of the first wave (t1) who could be reached were invited to attend a follow-up assessment (t2). Those participants who were able and willing to take part in the follow-up examination were sent a letter with multiple questions and questionnaires per mail. The GAD-7 was used for both the t1 and the t2 assessment. Both the baseline and the follow-up study have been approved by the Ethics Committee of the University of Leipzig (approval numbers 263-2009-14122009, 263/09-ff, and 201/17-ek). Written informed consent was obtained by all participants. Results of the baseline assessment of this study regarding the GAD-7 have already been published [7]. The present article further adds the results of the longitudinal analyses.

### Instruments

The GAD-7 [19] is a one-dimensional questionnaire designed to detect symptoms of generalized anxiety disorder according to the DSM-IV. The item scores range from 0 (not at all) to 3 (nearly every day), resulting in sum scores that range from 0 to 21. The GAD-2 is an ultra-short form of the GAD-7 [30] consisting of only two items. According to a recent study on sensitivity to change of the GAD-7 [23], we used change scores of 4 or greater to reflect a clinically important difference for individuals.

In addition to the GAD-7, the following instruments were included both at baseline and at follow-up: The Satisfaction With Life Scale SWLS [31] (general life satisfaction), the Short Form Health Survey–8 SF-8 [32] (quality of life), the Patient Health Questionnaire-15 PHQ-15 [33] (bodily complaints), the Life Orientation Test LOT-R [34] (dispositional optimism), and the ENRICHD Social Support Instrument [35] (social support).

Sociodemographic factors were obtained in a structured interview. Socioeconomic status (SES) was calculated in accordance with the Robert-Koch-Institute [36], integrating education, income, and professional position into one index. For the regression analyses, socioeconomic status was categorized into three strata.

### Statistical analysis

Mean score differences between two groups of participants were expressed with effect sizes (Cohen's $d$), relating the mean score differences to the pooled standard deviation. Cronbach's alpha coefficient was used to determine internal consistency. Coefficients of temporal stability were calculated with Pearson's correlation coefficients. Some researchers prefer to use intraclass correlations, however, most of the research on temporal stability in the literature uses

simple correlation coefficients. Thus, to enhance comparability with these studies, we also use these more common Pearson correlation coefficients.

To test the psychometric properties of the single items, we used the common discriminatory power coefficients that indicate the correlation between an item and the part-whole-corrected sum score. In addition to that, we performed discriminatory power analyses with the change scores. These coefficients indicate to what degree the change in a single item corresponds with the change of the scale after removing the item of interest.

For establishing measurement invariance, confirmatory factor analysis (CFA) models were estimated with the diagonally weighted least squares (DWLS) method with mean- and variance-adjusted test statistics. Model fit was judged using a combinational rule of comparative fit index (CFI) and standardized root mean square residual (SRMR) [37]. Based on this rule, poor fit was indicated if both CFI and SRMR exceed the threshold for acceptability, i.e., CFI < 0.95 and SRMR > 0.06. We also present the Tucker-Lewis index (TLI) and the root mean square error of approximation (RMSEA). Differences in model fit are expressed by the difference of CFI values (ΔCFI) between sequential models. A difference of at least 0.002 is considered a substantial change in model fit, and smaller differences are regarded as being negligible.

First, we tested the model for each time point (t1 and t2) separately. Then, an unconstrained model in which both time points were combined served as the baseline model. Acceptable fit of this model indicates configural invariance, i.e., the factor patterns remain constant over time (configural invariance). From this model, the detection of a violation of measurement invariance starts, using a forward approach. Parameters were constrained set by set, and released if necessary, in the following order: thresholds and weights (metric or weak invariance), then additionally intercepts (scalar or strong invariance), and finally residuals (full or strict invariance) [38].

If, as a result of constraining a set of parameters, the model fit decreased substantially compared to the model before, a search for partial invariance was executed, otherwise the next set of parameters was constrained. If the search for partial invariance identified a parameter that should not be constrained to equality between occasions, the constraint was released only if its release also lead to a substantial increase in model fit. If no further misspecified constraint could be identified, the next set of parameters were constrained. CFAs and measurement invariance analyses were calculated with R, version 4.1.1 [39] with the packages lavaan 0.6.9 and semTools 0.5.5 [40]. All other statistics were performed with SPSS version 27.

## Results

### Sample characteristics

Of the 10000 participants in the baseline examination, 9751 persons provided valid GAD-7 data. Characteristics of that sample have been published previously [7]. The response rate of the baseline examination was 33%. Using the criterion of at least six valid GAD-7 items at both t1 and t2, the final sample consisted of 5355 individuals, resulting in a final response rate of 18%. Sociodemographic characteristics of the participants are given in Table 1. The mean time interval between the t1 and the t2 examination was 6.04 years (SD = 0.42 years).

### Anxiety mean scores and item characteristics

Table 2 shows that anxiety increased during the 6-year period from 3.28 to 3.66. This difference ($d = 0.11$) is statistically significant with $p < 0.001$. Using the cut-off of 10 or higher for a heightened GAD-7 score [6], 90.8% of the sample had normal anxiety scores both at t1 and t2, 2.7% showed heightened anxiety only at t1, 4.7% only at t2, and 1.8% at both time points.

**Table 1. Sociodemographic characteristics of the final sample (*n* = 5355) at t1.**

| | Males | | Females | | Total sample | |
| | (*n* = 2509) | | (*n* = 2846) | | (*n* = 5355) | |
| | *n* | % | *n* | % | *n* | % |
|---|---|---|---|---|---|---|
| Age | | | | | | |
| Mean | 57.9 | | 56.8 | | 57.3 | |
| (SD) | (12.5) | | (12.2) | | (12.3) | |
| Age group | | | | | | |
| ≤ 39 years | 143 | 5.7 | 142 | 5.0 | 285 | 5.3 |
| 40–49 years | 576 | 23.0 | 755 | 26.5 | 1331 | 24.9 |
| 50–59 years | 541 | 21.6 | 649 | 22.8 | 1190 | 22.2 |
| 60–69 years | 701 | 27.9 | 782 | 27.5 | 1483 | 27.7 |
| ≥ 70 years | 548 | 21.8 | 518 | 18.2 | 1066 | 19.9 |
| Marital status [a] | | | | | | |
| Married, living together | 1719 | 68.6 | 1671 | 58.8 | 3390 | 63.4 |
| Married, living separately | 42 | 1.7 | 66 | 2.3 | 108 | 2.0 |
| Never married | 435 | 17.4 | 451 | 15.9 | 886 | 16.6 |
| Divorced | 252 | 10.1 | 411 | 14.5 | 663 | 12.4 |
| Widowed | 58 | 2.3 | 242 | 8.5 | 300 | 5.6 |
| Education [a] | | | | | | |
| < 10 years | 154 | 6.2 | 193 | 6.9 | 347 | 5.6 |
| 10–11 years | 1271 | 51.2 | 1613 | 57.3 | 2884 | 54.4 |
| ≥ 12 years | 1058 | 42.6 | 1011 | 35.9 | 2069 | 39.0 |
| Occupational status [a] | | | | | | |
| Working full time | 1275 | 51.2 | 1123 | 39.8 | 2398 | 45.2 |
| Working part-time | 80 | 3.2 | 417 | 14.8 | 497 | 9.4 |
| Unemployed | 114 | 4.6 | 125 | 4.4 | 239 | 4.5 |
| Retired | 985 | 39.6 | 1081 | 38.3 | 2066 | 38.9 |
| Other | 36 | 1.4 | 74 | 2.6 | 110 | 2.1 |
| Socio-economic status [a] | | | | | | |
| Low | 369 | 14.7 | 448 | 15.8 | 817 | 15.3 |
| Medium | 1467 | 58.6 | 1799 | 63.4 | 3266 | 61.1 |
| High | 668 | 26.7 | 590 | 20.8 | 1258 | 23.6 |

[a] Missing data not reported

The GAD-7 mean score of those individuals who participated at t1 but not at t2 (drop outs, *n* = 4396) was 3.91 ± 3.60, significantly higher ($p < 0.001$) than the t1 mean score of those who also attended the t2 examination (M = 3.28 ± 3.16).

Table 2 shows that all items except item I6 (being easily annoyed or irritable) contributed to the increase in anxiety from t1 to t2, with effect sizes between 0.03 and 0.18. The main contributors to this increase were items 1 (feeling nervous) ($d$ = 0.18) and item 7 (feeling afraid) ($d$ = 0.17).

All items contributed to the GAD-7 total score, both at t1 and t2, with discrimination power coefficients between 0.49 and 0.70. The column "discrimination power (Δitem, Δscale)" shows that all item changes from t1 to t2 contributed to the change of the GAD-7 total score with somewhat lower but nevertheless positive coefficients (between 0.36 and 0.53).

Table 2 also shows that the test-retest correlation of the GAD-7 sum score was 0.53, and that the test-retest correlations of the single items ranged from 0.34 to 0.45.

**Table 2. GAD-7 item and sum score characteristics.**

| Item | t1 | | t2 | | Difference t2-t1 | | ES t2-t1 | discr. power t1 | discr. power t2 | discr. power (Δitem, Δscale) | $r_{tt}$ |
|---|---|---|---|---|---|---|---|---|---|---|---|
| | M | (SD) | M | (SD) | M | (SD) | | | | | |
| 1. Feeling nervous | .49 | (.66) | .61 | (.69) | .12 | (.73) | .18 | .65 | .71 | .52 | .41 |
| 2. Not able to stop worrying | .34 | (.59) | .39 | (.62) | .05 | (.69) | .08 | .66 | .70 | .53 | .35 |
| 3. Worry about different things | .52 | (.65) | .54 | (.67) | .02 | (.76) | .03 | .65 | .67 | .52 | .34 |
| 4. Trouble relaxing | .65 | (.71) | .69 | (.76) | .03 | (.77) | .05 | .63 | .67 | .49 | .45 |
| 5. Being restless | .41 | (.66) | .46 | (.71) | .05 | (.77) | .07 | .49 | .51 | .36 | .37 |
| 6. Easily annoyed or irritable | .60 | (.62) | .60 | (.62) | .01 | (.69) | .00 | .49 | .51 | .38 | .37 |
| 7. Feeling afraid | .26 | (.53) | .36 | (.63) | .10 | (.65) | .17 | .59 | .62 | .46 | .37 |
| GAD-2 sum score | 0.83 | (1.10) | 1.00 | (1.18) | .17 | (1.20) | .15 | α = .71 | α = .75 | α = .58 | .44 |
| GAD-7 sum score | 3.28 | (3.16) | 3.66 | (3.46) | .38 | (3.21) | .11 | α = .84 | α = .86 | α = .75 | .53 |

Note: M: mean; SD: standard deviation; ES: Effect size of the difference t2 score minus t1 score; discr. power; discrimination power; Δ: difference t2 score minus t1 score; $r_{tt}$: test-retest correlation; α: Cronbach's alpha

The correlation between the GAD-7 scores and the GAD-2 scores were 0.87 (at t1) and 0.88 (at t2), and the correlation between the GAD-7 change (t2 minus t1) score and the GAD-2 change score was 0.80.

## Measurement invariance

The results of the measurement invariance analyses are presented in Table 3. When t1 and t2 were analyzed separately, CFA results indicated a good model fit for both measurement points. Taken both measurement points together in one model, full invariance could be established.

## The impact of sociodemographic variables on the course of anxiety

Table 4 presents mean score differences between certain groups of participants. Females were more anxious than males at t1 and at t2, and the increase in anxiety from t1 to t2 was slightly and insignificantly higher in females (Δ = 0.40) than in males (Δ = 0.35). All age groups except for the age group 60–69 years showed an increase in anxiety. While SES was negatively associated with anxiety in the cross-sectional perspective, there was no such linear relationship for the changes in anxiety; the difference scores were between 0.26 and 0.42 for the three SES groups.

Based on the criterion of a clinically meaningful change in anxiety being four or more points [23], 428 participants (8.0%) showed relevant decrease in anxiety, 4249 (79.3%) showed

**Table 3. Testing for measurement invariance across t1 and t2.**

| | NPar | Chi$^2$ (df) | Chi$^2$/df | CFI | SRMR | TLI | RMSEA | ΔCFI | ΔSRMR | ΔTLI | ΔRMSEA |
|---|---|---|---|---|---|---|---|---|---|---|---|
| t1 | 28 | 457.9 (14) | 32.7 | 0.983 | 0.039 | 0.975 | 0.077 | . | . | . | . |
| t2 | 28 | 646.8 (14) | 46.2 | 0.982 | 0.045 | 0.972 | 0.092 | . | . | . | . |
| Config. invariance | 64 | 1043.8 (69) | 15.1 | 0.983 | 0.035 | 0.977 | 0.051 | . | . | . | . |
| Metric invariance | 51 | 882.7 (82) | 10.8 | 0.986 | 0.036 | 0.984 | 0.043 | 0.003 | 0.001 | 0.007 | -0.008 |
| Scalar invariance | 45 | 993.9 (88) | 11.3 | 0.984 | 0.036 | 0.984 | 0.044 | 0.002 | <0.001 | <0.001 | 0.001 |
| Full invariance | 38 | 1036.3 (95) | 10.9 | 0.983 | 0.036 | 0.984 | 0.043 | 0.001 | <0.001 | <0.001 | -0.001 |

Note. NPar: number of parameters; Chi$^2$: scaled chi-squared statistic; df: degrees of freedom; CFI: scaled comparative fit index, SRMR: standardized root mean square residual; TLI: scaled Tucker-Lewis index; RMSEA: scaled root mean square error of approximation; Δ: difference of fit indices between sequential (nested) models.

**Table 4. Changes in anxiety, broken down by sex, age group, and SES group.**

|  | t1 | | t2 | | Diff. t2-t1 | | Change [a] | | | $r_{tt}$ |
|---|---|---|---|---|---|---|---|---|---|---|
|  | M | (SD) | M | (SD) | M | (SD) | de-crease (%) | no change (%) | in-crease (%) |  |
| **Sex** |  |  |  |  |  |  |  |  |  |  |
| Males | 2.72 | (2.87) | 3.07 | (3.15) | 0.35 | (2.93) | 6.5 | 83.1 | 10.4 | .53 |
| Females | 3.77 | (3.32) | 4.18 | (3.64) | 0.40 | (3.44) | 9.3 | 76.0 | 14.4 | .52 |
| Age |  |  |  |  |  |  |  |  |  |  |
| ≤ 39 y. | 3.14 | (2.75) | 3.95 | (3.46) | 0.81 | (3.31) | 6.3 | 77.9 | 15.8 | .45 |
| 40–49 y. | 3.42 | (3.34) | 3.98 | (3.72) | 0.55 | (3.45) | 7.5 | 78.4 | 14.1 | .53 |
| 50–59 y. | 3.55 | (3.39) | 3.86 | (3.70) | 0.31 | (3.29) | 9.1 | 77.7 | 13.2 | .57 |
| 60–69 y. | 3.07 | (3.05) | 3.06 | (2.99) | -0.01 | (2.91) | 8.7 | 82.5 | 8.8 | .54 |
| ≥ 70 y. | 3.13 | (2.87) | 3.79 | (3.39) | 0.66 | (3.11) | 6.8 | 78.4 | 14.7 | .52 |
| SES |  |  |  |  |  |  |  |  |  |  |
| low | 3.82 | (3.58) | 4.19 | (3.83) | 0.37 | (3.56) | 9.9 | 75.6 | 14.4 | .54 |
| medium | 3.29 | (3.13) | 3.71 | (3.49) | 0.42 | (3.19) | 7.8 | 79.1 | 13.1 | .54 |
| high | 2.91 | (2.89) | 3.17 | (3.09) | 0.26 | (3.00) | 7.3 | 82.4 | 10.3 | .50 |
| All | 3.28 | (3.16) | 3.66 | (3.46) | 0.38 | (3.21) | 8.0 | 79.3 | 12.7 | .53 |

$r_{tt}$: test-retest correlation; [a]: changes of at least four points

no relevant change, and 678 (12.7%) showed a relevant increase in anxiety. The corresponding proportions, broken down by sex, age group, and SES group, are also given in Table 4.

Regarding the temporal stability $r_{tt}$, Table 4 shows that there were nearly no sex differences (coefficients of 0.53 and 0.52 for males and females, respectively). Concerning age, the highest stability was found for the age group 50–59 years ($r_{tt} = 0.57$).

## Correlations with other psychological or QoL variables

The highest cross-sectional correlations of the GAD-7 were found for the mental health component of the SF-8 ($r = 0.68$ at t1 and $r = 0.66$ at t2). The comparison between the correlations at t1 with those at t2 indicates only small differences between the measurement points with the exception of the LOT-R correlations, which were somewhat higher at t2.

The last column of Table 5 presents the (longitudinal) correlations between the changes (increases or decreases from t1 to t2) of the GAD-7 score with the changes of the other scales. All of these correlations are smaller than the corresponding cross-sectional correlations, but the sequence of the correlations is very similar to that of the raw scores: Variables with high cross-sectional associations such as Mental Health also show relatively high associations between the change scores. To illustrate the association between GAD-7 change and the change in the other variables using the PHQ-15 as an example, we calculated the mean change scores of the PHQ-15 (t2 score minus t1 score) for each of the three GAD-7 change categories. The results were: ΔPHQ-15 = -1.12 ± 3.27 for the group with meaningful decrease in anxiety, ΔPHQ-15 = 0.44 ± 2.91 for the group with no meaningful change in anxiety, and ΔPHQ-15 = 2.47 ± 3.77 for the group with meaningful increases in anxiety from t1 to t2.

## Discussion

The first aim of this study was to examine whether anxiety levels change over a 6-year period. While the small age differences in anxiety suggest that an additional six life-years should result in only marginal mean score changes, the t2 mean scores were nevertheless higher than those of the t1 measurement ($d = 0.11$). When comparing the mean values of t1 and t2, two

**Table 5. Correlations between the GAD-7 scores and other variables.**

|  | r (GAD-7, scale) at t1 | r (GAD-7, scale) at t2 | r (Δ GAD-7, Δ scale) |
|---|---|---|---|
| SWLS: Life satisfaction | -.41 | -.46 | -.30 |
| ENRICHD: Social Support | -.26 | -.30 | -.14 |
| LOT-R Optimism | -.30 | -.40 | -.18 |
| LOT-R Pessimism | .28 | .37 | -.16 |
| PHQ-15: Complaints | .52 | .54 | .31 |
| SF-8: Physical functioning | -.27 | -.27 | -.13 |
| SF-8: Role-physical | -.33 | -.33 | -.17 |
| SF-8: Bodily pain | -.29 | -.31 | -.10 |
| SF-8: General health | -.39 | -.36 | -.22 |
| SF-8: Vitality | -.39 | -.39 | -.21 |
| SF-8: Social functioning | -.50 | -.48 | -.30 |
| SF-8: Role-emotional | -.52 | -.51 | -.33 |
| SF-8: Mental health | -.69 | -.66 | -.49 |
| SF-8: Physical Component Summmary | -.24 | -.24 | -.07 |
| SF-8: Mental Component Summary | -.68 | -.66 | -.48 |

differences should be considered. Firstly, the t1 examination was carried out at the study center, while at t2 the questionnaire was completed at home, and the completed t2 questionnaire was sent to the study center by mail. Second, the end of the t2 study period coincided with the beginning of the COVID-19 pandemic, and this may have increased anxiety levels in the general population. However, a systematic review and meta-analysis comparing mental health before versus during the COVID-19 pandemic in 2020 [41] found that the anxiety mean scores in the older adult general population increased only slightly or not at all due to the COVID-19 pandemic, a result that was also confirmed by further general population studies [42–44]. Therefore, we do not think that the partial overlap with the COVID-19 pandemic had a substantial impact on the t2 results, though we cannot quantify this possible effect. In addition, there is no evidence that completing questionnaires during a laboratory study would generally overestimate or underestimate results compared with a postal survey. Thus, the results of this study suggest that anxiety levels really do increase over time. A similar result was also found in a previous study using the SF-36, that found out deteriorations in mental health over a 6-year period [45], though such age effects could not be detected when analyzed in a cross sectional study [46]. This also means that cross-sectional studies with comparisons of age groups may be insufficient tools for predicting changes in QoL and mental health variables over time.

The GAD-7 proved to be a reliable instrument. While psychometric characteristics designed for cross-sectional studies have already been established in multiple studies, the current investigation adds that the items are also appropriate for detecting long-term changes, with coefficients of at least 0.36 for the association between item change and part-whole corrected scale change. Since this way of calculating coefficients for the reliability of change has not been applied to the GAD-7 before, it is not possible to compare the results with other investigations.

Measurement invariance was also established from the long–term perspective. While such measurement invariance has already been found in patient groups [27, 28], this study adds that the longitudinal comparisons of GAD-7 examinations are also justified in the general population. Though there were differences between the t1 and the t2 examination concerning setting and possible partial impact of the COVID-19 pandemic, the results of the measurement invariance analyses show that the comparisons between the t1 and the t2 scores are fair.

Beyond the mean score changes, the correlations between the t1 and the t2 scores indicate the degree of temporal stability of anxiety. Our results ($r = 0.53$ for the total sample) are in line with findings from the literature, where the following stability coefficients for different anxiety questionnaires and different time intervals are reported: $r = 0.59$ (3 years) [47], $r = 0.60$ (4 years) [48], $r = 0.50$ (5 years) [49], and $r = 0.55$ (6 years) [50]. A study with a 3-months time interval found a higher stability coefficient of $r = 0.65$ [51], however, it is plausible that the stability decreases with increasing amounts of time elapsed between measurements [11, 48].

What this study also adds is that there are no sex differences in the temporal stability of anxiety ($r = 0.52$ for males and $r = 0.53$ for females). Though women tend to report more anxiety and more emotional instability than men, their degree of fluctuation in this 6-year period is not higher than that of males. Regarding age, there were no clear age effects on the temporal stability. The lowest coefficient ($r = 0.45$) was observed for the youngest age group ($\leq 39$ years) which, however, should not be over-interpreted because of the relatively small sample size of that group. It is remarkable that despite the high increase in anxiety in the oldest age group, the predictability ($r = 0.52$) is nearly as high as that of the total sample. The consequence for the interpretation of long-term stability examinations in older patient groups is that increases in anxiety are common, but that the individual predictability of the future state of anxiety is not affected by old age. Though life changes and losses in older age (increasing health problems; loss loved ones) lead to a slight increase in the anxiety mean score level, this increase does not show more individual differences in the older group than in the other groups.

Living without a partner and being unemployed are associated with higher levels of anxiety [7]. However, our results show that not having a partner or being unemployed does not predict changes in anxiety. That is, the difference in anxiety between employed and unemployed people neither increases nor decreases on a mean score level.

The relationship between anxiety and other QoL or mental health variables has already been examined in multiple studies. Our new analysis adds associations observable over the long-term perspective. Changes in anxiety were correlated with changes in the other variables, though the correlations of the change scores were lower than the correlations of the raw scores in the cross-sectional design, a result that has also been reported elsewhere [18]. The highest change score correlations were those that also showed high cross-sectional correlations (e.g., correlations with the Mental health scale of the SF-8). Research projects that aim at distinguishing between certain types of clinical or health variables (e.g. fatigue and depression) can profit from such longitudinal correlation profiles.

## Limitations

The response rate of the study was low (33% at baseline). Persons with poor mental health were probably underrepresented in the baseline assessment; a comparison between the 10000 participants of the t1 assessment and those non-participants who were nevertheless willing to answer several questions concerning their behavior and health state [52] found several differences between these groups. Compared with non-participants, participants in the study had the following characteristics: higher education, higher proportion of married and employed individuals, more non-smokers, and better physical health, e.g., with respect to cardiovascular disease and diabetes [52]. Moreover, an analysis of the overall survival rate of the study participants showed that their survival rate was higher than the German average [53]. Less than 60% of the t1 participants took part in the t2 examination, and we do not know the course of anxiety in those who dropped out. We also do not have information on reasons for non-participation at t2. Possible reasons are death, being too ill to take part in the t2 examination, and loss of interest after having obtained some health-related information already at t1, lack of time, or

having moved. Such limitations in representativeness are common in epidemiological research. Nevertheless, they show that the mean scores of the t1 and the t2 examination should be interpreted with caution.

## Conclusions

The results of this study provide a deeper insight into the conditions that are relevant for analyses of change in anxiety over several years. The GAD-7 proved to be a reliable instrument for longitudinal studies.

## Acknowledgments

Leipzig Research Centre for Civilization Diseases (LIFE) is an organizational unit affiliated to the Medical Faculty of the University Leipzig. The authors thank the participants of the LIFE-Adult-Study for taking part in the study and the complete LIFE-Adult team for organizing the course of the study and performing the examinations.

## Author Contributions

**Conceptualization:** Andreas Hinz, Heide Glaesmer.

**Data curation:** Michael Friedrich.

**Formal analysis:** Andreas Hinz, Michael Friedrich.

**Investigation:** Anja Mehnert-Theuerkauf.

**Writing – original draft:** Andreas Hinz, Katja Petrowski, Anne Toussaint.

**Writing – review & editing:** Andreas Hinz, Peter Esser, Heide Glaesmer, Anja Mehnert-Theuerkauf, Matthias L. Schroeter.

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
