## [Decision Letter · Decision Letter 0]

29 May 2023

PONE-D-23-01688Changes in anxiety in the general population over a six-year periodPLOS ONE

Dear Dr. Hinz,

Thank you for submitting your manuscript to PLOS ONE. After careful consideration, we feel that it has merit but does not fully meet PLOS ONE’s publication criteria as it currently stands. Therefore, we invite you to submit a revised version of the manuscript that addresses the points raised during the review process.

We look forward to receiving your revised manuscript.

Kind regards,

Md. Saiful Islam, BPH, MPH

Academic Editor

PLOS ONE

Journal Requirements:

-https://www.sciencedirect.com/science/article/abs/pii/S0022399922002999?via%3Dihub

In your revision ensure you cite all your sources (including your own works), and quote or rephrase any duplicated text outside the methods section. Further consideration is dependent on these concerns being addressed."

Reviewers' comments:

Reviewer's Responses to Questions

**Comments to the Author**

1. Is the manuscript technically sound, and do the data support the conclusions?

Reviewer #1: Yes

Reviewer #2: Yes

Reviewer #3: Yes

2. Has the statistical analysis been performed appropriately and rigorously? 

Reviewer #1: Yes

Reviewer #2: No

Reviewer #3: Yes

3. Have the authors made all data underlying the findings in their manuscript fully available?

Reviewer #1: Yes

Reviewer #2: Yes

Reviewer #3: Yes

4. Is the manuscript presented in an intelligible fashion and written in standard English?

Reviewer #1: Yes

Reviewer #2: Yes

Reviewer #3: Yes

5. Review Comments to the Author

Reviewer #1: The sample is not representative of anything.

There is an educational bias.

The different survey methods of the two surveys have not been discussed. Invariance analyses are insufficient there.

The educational grouping in Table 1 is ambiguous.

The influence of Covid-19 in the second survey is not discussed.

Reviewer #2: The author Andreas Hinz et al “Changes in anxiety in the general population over a six-year period is a well-written and informative article that addresses important research questions related to anxiety in the general population.

Major Comments:

1) The authors could please highlight about the sample size and characteristics are not well described which could limit the findings.

2) The study sample size is relatively small, final sample of only 5355 individuals. The author need to address this finding as this lead to issues with statistical power.

3) The response rate of baseline examination was only 33%, the author should provide additional information on the characteristics of those who did not respond and how they may differ from those who did.

4) GAD-7 score of those who dropped out was significantly higher than those who attended both examinations could introduce bias into the study. The author should explore potential reasons for the differential dropout rates and assess whether this could impact the validity of their findings.

Reviewer #3: Thank you for inviting me to review this study entitled “Changes in anxiety in the general population over a six-year period”

General comment: I appreciate the authors' efforts in this study. This manuscript is written well. However, I request the authors to consider the following suggestions for further improvement of the manuscript.

The references are also quite old more than 10 years accounting for about 45% of the reference list, and should not exceed more than 20%.

Minor Comments

Comments # 1: Abstract

Methods: Please insert single backspace for the lines “The GAD-7 was included in an examination with two waves, six years apart”.

Please include the standard deviation (SD) with the population mean age.

Conclusion: what is the recommendation of this study? Please highlights a single line in the conclusion section.

Comments # 2: Introduction

In the 3rd para, however, from a longitudinal perspective, it is also of interest whether changes in anxiety correspond with changes in those variables. The word changes are inserted in different fonts; please maintain the same fonts for this word.

Comments # 3: Methods

The study protocol focused on the age range of 40-80, but the sample selection ranged from 18-80. What are the reasons for including the age range of 18-80 and the exclusion criteria of the study participants?

The first wave of this study was conducted between 2011 and 2014, and The second wave was conducted between 2017 and 2021. Why was the time interval not the same between the two waves? I can’t understand the reasons; Please clear it up.

What is the ethical approval reference number? Please included this.

6. PLOS authors have the option to publish the peer review history of their article (what does this mean?). If published, this will include your full peer review and any attached files.

Reviewer #1: No

Reviewer #2: **Yes: **Naresh Poondla

Reviewer #3: No

---

## [Author Response · Author response to Decision Letter 0]

25 Jul 2023

We thank the editor and the reviewers very much for their time, their careful reading and their helpful comments.

Journal Requirements:

RESPONSE.

We changed the format according to the requirements and hope that it is correct now.

-https://www.sciencedirect.com/science/article/abs/pii/S0022399922002999?via%3Dihub

 In your revision ensure you cite all your sources (including your own works), and quote or rephrase any duplicated text outside the methods section. Further consideration is dependent on these concerns being addressed."

RESPONSE.

There was a small overlap in the description of the Methods since both texts are based on the same examination. We found only one identical sentence and changed it accordingly.

RESPONSE.

We now included full ethics statement in the Methods section. 

Reviewer #1: 

The sample is not representative of anything.

RESPONSE.

The study was designed to include a representative sample of the population of the city of Leipzig in the age range 40 to 80. This has largely been achieved, with representativeness referring to age, gender and city district. In addition, (relatively fewer) people under the age of 40 were included in the study for comparison purposes. We now have the option of either limiting ourselves to those over 40 years of age, or including younger individuals as well. From our point of view, it is useful to include the younger ones as well, in order to be able to represent the age dependency of the anxiety scores in a larger age range.

The more serious problem with representativeness is that in such epidemiological studies persons of lower social classes and persons with poor physical and mental health are underrepresented. Because of the voluntary nature of the study, this cannot be avoided entirely. This point is now discussed in more detail in the Limitations section.

Fortunately, we have some information on the non-participants that give an idea of the degree of non-representativeness (see Limitations section).

There is an educational bias.

RESPONSE.

In our study, individuals with lower educational attainment are actually underrepresented. This is a problem that occurs in most epidemiological studies. The problem of non-representativeness is now mentioned in the Limitations section.

The different survey methods of the two surveys have not been discussed. Invariance analyses are insufficient there.

RESPONSE.

We agree with the reviewer that invariance analyses are not sufficient for demonstrating an equivalence of the t1 and t2 assessments. When comparing t1 values with t2 values, it should be noted that not only were participants six years older, but there were also differences in setting (study center survey at t1 and postal survey at t2), and that the t2 survey was partially into the onset of the Covid-19 pandemic. However, the results of the measurement invariance analysis show that even despite these differences, measurement invariance largely exists. We believe that such a result demonstrates the robustness of the GAD-7 scale structure. The problem of different settings and possible influences of Covid-19 is now also addressed in the Discussion section. 

The educational grouping in Table 1 is ambiguous.

RESPONSE.

We apologize for the mistake and corrected the table accordingly. 

The influence of Covid-19 in the second survey is not discussed.

RESPONSE.

In the Discussion section we now discuss the potential impact of the Covid-19 pandemic. 

Reviewer #2: 

The author Andreas Hinz et al “Changes in anxiety in the general population over a six-year period is a well-written and informative article that addresses important research questions related to anxiety in the general population.

Major Comments:

1) The authors could please highlight about the sample size and characteristics are not well described which could limit the findings.

RESPONSE.

In Table 1, we have described the main characteristics of the sample. It would be possible to add further characteristics. However, we think that for the topic of this manuscript the details of the sample description are sufficient. Problems of possible non-representativeness of the present sample are now discussed in detail in the discussion section.

2) The study sample size is relatively small, final sample of only 5355 individuals. The author need to address this finding as this lead to issues with statistical power.

RESPONSE.

We believe that a sample size of more than 5000 people is relatively large for a longitudinal study. Nevertheless, once more we acknowledge the problem of representativeness. In the Discussion section we addressed this issue further.

3) The response rate of baseline examination was only 33%, the author should provide additional information on the characteristics of those who did not respond and how they may differ from those who did.

RESPONSE.

In the Discussion section we already mentioned that the non-participants were less healthy than the participants. Fortunately, we have some data about the non-participants, and we enlarged the description of the non-participants in more detail.

4) GAD-7 score of those who dropped out was significantly higher than those who attended both examinations could introduce bias into the study. The author should explore potential reasons for the differential dropout rates and assess whether this could impact the validity of their findings.

RESPONSE.

Unfortunately, we do not have data on reasons for non-participation in the second wave. Potential reasons are now given in the Discussion section.

Reviewer #3: 

Thank you for inviting me to review this study entitled “Changes in anxiety in the general population over a six-year period”

General comment: I appreciate the authors' efforts in this study. This manuscript is written well. However, I request the authors to consider the following suggestions for further improvement of the manuscript.

The references are also quite old more than 10 years accounting for about 45% of the reference list, and should not exceed more than 20%.

RESPONSE.

In the first version the proportion of references more than 10 years old were 22/55 = 40%. In the revision, we replaced several references by newer ones. However, for the references to the questionnaires in the Methods section it seems to be appropriate to refer to the original publications, which are at least 10 or 20 years old in most cases. 

In the revised version we now have a proportion of 16/53 = 30% of references before 2013. 

Minor Comments

Comments # 1: Abstract

Methods: Please insert single backspace for the lines “The GAD-7 was included in an examination with two waves, six years apart”.

RESPONSE.

We included the backspace.

Please include the standard deviation (SD) with the population mean age.

RESPONSE.

We also included the SD in the abstract.

Conclusion: what is the recommendation of this study? Please highlights a single line in the conclusion section.

RESPONSE.

In the present form, the abstract has two conclusions (and, therefore, recommendations): the appropriateness of the instrument, and the fact that age and sex are not relevant when interpreting change scores. 

We are not quite sure what is meant with “highlights a single line”. Should we restrict to one of these two conclusions, or should we write a new conclusion? Since we believe that both conclusions are relevant, we would like to maintain that Conclusion part. 

Comments # 2: Introduction

In the 3rd para, however, from a longitudinal perspective, it is also of interest whether changes in anxiety correspond with changes in those variables. The word changes are inserted in different fonts; please maintain the same fonts for this word.

RESPONSE.

We unified the fonts in the word “changes”.

Comments # 3: Methods

The study protocol focused on the age range of 40-80, but the sample selection ranged from 18-80. What are the reasons for including the age range of 18-80 and the exclusion criteria of the study participants?

RESPONSE.

There were two options for analyzing the data set: (a) restricting the analysis to persons aged 40 and older, or including the (smaller) subsample of persons under 40. Both options have advantages and disadvantages. We think that in the context of the present study on the course of anxiety scores, it is also interesting to report the results of the under-40s. Although the number of younger persons (n=285) is smaller compared to the older ones, the number of cases still seemed sufficient to us to report the results.

The first wave of this study was conducted between 2011 and 2014, and The second wave was conducted between 2017 and 2021. Why was the time interval not the same between the two waves? I can’t understand the reasons; Please clear it up.

RESPONSE.

The second wave of the study lasted a little longer than the first wave. This was also due to the fact that those individuals who did not initially respond to the invitation to wave 2 were still sent reminder letters, and that the time window for responses to the t2 wave was extended somewhat in order to increase the response rate even more.

What is the ethical approval reference number? Please included this.

RESPONSE.

We included the reference number of the approval of the ethics committee.

---

## [Decision Letter · Decision Letter 1]

24 Aug 2023

Changes in anxiety in the general population over a six-year period

PONE-D-23-01688R1

Dear Dr. Andreas Hinz,

We’re pleased to inform you that your manuscript has been judged scientifically suitable for publication and will be formally accepted for publication once it meets all outstanding technical requirements.

Kind regards,

Md. Saiful Islam, BPH, MPH

Academic Editor

PLOS ONE

Reviewers' comments:

Reviewer's Responses to Questions

**Comments to the Author**

1. If the authors have adequately addressed your comments raised in a previous round of review and you feel that this manuscript is now acceptable for publication, you may indicate that here to bypass the “Comments to the Author” section, enter your conflict of interest statement in the “Confidential to Editor” section, and submit your "Accept" recommendation.

Reviewer #2: All comments have been addressed

2. Is the manuscript technically sound, and do the data support the conclusions?

Reviewer #2: Yes

3. Has the statistical analysis been performed appropriately and rigorously? 

Reviewer #2: Yes

4. Have the authors made all data underlying the findings in their manuscript fully available?

Reviewer #2: Yes

5. Is the manuscript presented in an intelligible fashion and written in standard English?

Reviewer #2: Yes

6. Review Comments to the Author

Reviewer #2: Author has provided reviewer comments. Thank you for providing the comments as per the reviewers comments.

7. PLOS authors have the option to publish the peer review history of their article (what does this mean?). If published, this will include your full peer review and any attached files.

Reviewer #2: **Yes: **Naresh Poondla

---

## [Editor Report · Acceptance letter]

4 Sep 2023

PONE-D-23-01688R1 

Changes in anxiety in the general population over a six-year period 

Dear Dr. Hinz:

I'm pleased to inform you that your manuscript has been deemed suitable for publication in PLOS ONE. Congratulations! Your manuscript is now with our production department. 

Kind regards, 

on behalf of

Dr. Md. Saiful Islam 

Academic Editor

PLOS ONE